# Single-Step Fabrication of a Dual-Sensitive Chitosan Hydrogel by C-Mannich Reaction: Synthesis, Physicochemical Properties, and Screening of its Cu$^{2+}$ Uptake

**John Rey Apostol Romal [1,2,\*]** and **Say Kee Ong [2]**

1   Department of Chemistry, Grand View University, Des Moines, IA 50316, USA
2   Department of Civil, Construction, and Environmental Engineering, Iowa State University, Ames, IA 50011, USA
\*   Correspondence: jromal11@iastate.edu

**Abstract:** Uncovering the value of waste materials is one of the keys to sustainability. In this current work, valorization of chitosan was pursued to fabricate a novel modified chitosan functional hydrogel using a process-efficient protocol. The fabrication proceeds by a one-pot and single-step C-Mannich condensation of chitosan (3% $w/v$), glutaraldehyde (20 eq.), and 4-hydroxycoumarin (40 eq.) at 22 °C in 3% $v/v$ acetic acid. The Mannich base modified chitosan hydrogel (CS-MB) exhibits a dual-responsive swelling behavior in response to pH and temperature that has not been observed in any other hydrogel systems. Combining the pre-defined optimal swelling pH (pH = 4) and temperature (T = 22 °C), the CS-MB was screened for its Cu$^{2+}$ adsorption capacity at this condition. The CS-MB achieved an optimal adsorption capacity of 12.0 mg/g with 1.2 g/L adsorbent dosage after 36 h with agitation. The adsorption of Cu$^{2+}$ on the surface of CS-MB was verified by EDS, and an overview of the adsorption sites was exhibited by FT-IR. The simply fabricated novel CS-MB hydrogel under investigation presents a unique response to external stimuli that exhibits a promise in heavy metal removal from aqueous media.

**Keywords:** chitosan; hydrogel; dual responsive; Mannich reaction; water treatment; environmental engineering; heavy metal removal

## 1. Introduction

Chitin is a natural biopolymer extracted from some fungi, crustaceans, and bivalve shells that can be deacetylated to obtain its reactive derivative chitosan. Chitosan contains reactive substituents that are often subjected to a variety of chemical functionalization, including the production of functional hydrogels, and have been more commonly utilized in biomedical applications in the past [1], attracting more attention in water treatment applications more recently [2–4]. The natural properties of chitosan hydrogel of low solubility in aqueous system and poor mechanical strength, however, generally present drawbacks on its deployment as a bare chitosan functional hydrogel system [5,6]. Fabrication strategies have become available to overcome this limitation including chemical crosslinking of chitosan with common crosslinking agents such as genepin [7], epichlorohydrin [8], and mono/dialdehydes [9,10], to name a few, to improve control over the crosslinked-chitosan's mechanical and physicochemical properties.

One of the most implemented chemical crosslinking strategies for the fabrication of chitosan functional hydrogel is by employing glutaraldehyde. This is partly because glutaraldehyde, among others, is cheap and commercially abundant, is known to be a highly reactive crosslinker for amine-containing biomolecules, and can establish a covalent crosslinking bridge between biomolecule or polysaccharide backbones [11]. In biomedical applications, glutaraldehyde crosslinked chitosan hydrogel impregnated with

diclofenac sodium has shown a controlled release of the drug [12]. The same glutaraldehyde crosslinked chitosan hydrogel system has also been doped with calcium phosphate to improve biomechanical properties and bone tissue biocompatibility [13]. Furthermore, in the perspective of water treatment application, the glutaraldehyde–chitosan hydrogel offers an improved ion permeability and water absorption [14] and composites of this hydrogel system present enhanced adsorption capability and selectivity for aqueous pollutants. To highlight its growing utilization in water treatment, composites of glutaraldehyde–chitosan hydrogel has shown to remove proteins [15], radioactive metals [16], heavy metals [17], N-containing contaminants [18], and small pharmaceutical pollutants [19] in aqueous environments. Considering the on-going environmental concerns over contamination in water systems, this current study, although mainly focusing on the fabrication method development and characterization, will also survey the activity of a modified chitosan hydrogel as an adsorbent for the removal of $Cu^{2+}$ from an aqueous medium.

Currently, there is still no unanimity to a singular crosslinking mechanism of glutaraldehyde–chitosan hydrogel system, as both chitosan and glutaraldehyde, on their own, can be in different forms (cationic vs. anionic, monomeric vs. oligomeric, etc.) at different conditions [20]. The more widely accepted crosslinking mechanism of a glutaraldehyde–chitosan hydrogel system that is reported in many more recent works is through a Schiff base reaction that is connected by an imine linkage from either or both ends of the dialdehyde and amine of the chitosan backbone [21–23]. Crosslinking by Schiff base, however, has its consequence in terms of chemical integrity in a way that the imine functional group is susceptible to hydrolysis under acidic environment that may cause the glutaraldehyde–chitosan hydrogel system to collapse, which is deemed to be impractical in water treatment application. Successful studies reported on metal removal from water using a glutaraldehyde–chitosan hydrogel systems are those that are composites of inorganic supports such as titanium oxide [17] and silica/$Fe_3O_4$ [24] or grafted with organic moieties such as graphene oxide [25], N-aminorhodamine [26], methionine [27], and melamine [28]. It is speculated that compositing, grafting, and doping strategies for glutaraldehyde–chitosan hydrogel systems are essential to synergistically improve the chemical stability of the imine moiety in the glutaraldehyde–chitosan hydrogel, avoiding collapse in their system under detrimental conditions.

Although the glutaraldehyde crosslinked chitosan hydrogel composites provide chemically stable adsorbent materials with excellent metal-adsorption capacities, fabrication of these hydrogel composites, especially those composites of transition metal oxides, in large scale can be material costly. Moreover, grafting organic moieties onto the glutaraldehyde–chitosan hydrogel system is usually fabricated in multi-step and non-ambient conditions of synthetic methodology that collectively generates energy costs and accumulates wastes, which are also not feasible for large-scale production. To address cost-inefficient and complex methods to fabricate a functional hydrogel of chitosan for the removal of contaminants from aqueous systems, this current work explores a facile methodology to fabricate a modified chitosan hydrogel system crosslinked by a beta-amino ketone (CS-MB) through a three-component C-Mannich reaction of chitosan (CS), glutaraldehyde (Glu), and 4-hydroxycoumarin (4-HC). As opposed to the susceptibility of imine linkages to hydrolyze under acidic conditions, beta-amino ketones were reported to withstand elimination under these conditions [29,30], which is a crucial characteristic for a material employed in water treatment. Beta-amino ketone was previously utilized as a metal chelator for the recovery of metals from metal salts by chemical vapor deposition [31]. Additionally, the CS-MB contains a lactone moiety from the 4-HC that was previously reported to complex with heavy metals [32]. A specific chemical characteristic to the coumarin moiety is the potential synergistic effect of the highly conjugated system that participates in both coordination, chelation, and non-covalent interactions by a cation–$\pi$ interaction [33], which can interact with metal cations at optimal conditions, adding to the motivation to pursue this current study. To the best of the authors' knowledge, the fabrication method reported in this work is by far the first to generate a beta-aminoketone crosslinked chitosan-based hydro-

gel with a unique dual-responsive swelling characteristic by a one-pot, single-step, and ambient-condition C-Mannich reaction.

## 2. Materials and Methods

### 2.1. Reagents

Low molecular weight CS (50–190 kDa and 75–85% deacetylation degree), 4-hydroxyc= oumarin (97% purity), and glacial acetic acid (ACS reagent) were purchased from Sigma-Aldrich, Co, St. Louis, MO 63103, USA. Glutaraldehyde (50% aq. solution) was purchased from Thermo Scientific. Anhydrous ethanol (200 proof) was purchased from Decon Labs, Inc., King of Prussia, PA 19406, USA. Copper chloride dihydrate was purchased from Sigma-Aldrich, Co. Distilled water and nanopure water was prepared in-house.

### 2.2. General Method for CS-MB Hydrogels Fabrication

A homogenous solution of 3.0% $w/v$ of CS (625 mg, 1 eq.) was prepared with an aqueous solution of 3% $v/v$ acetic acid (20.5 mL) under magnetic stirring at 22 °C for 30 min. A mixture of 50% glutaraldehyde aq. solution (0.25 mmol, 50 µL, 20 eq.) and 4-HC (0.50 mmol, 81 mg, 40 eq.) in 5 mL linker solution (8:2 of ethanol to 3% $v/v$ acetic acid) was prepared and added to the CS solution using a plastic syringe. The mixture was magnetically stirred at 22 °C and open atmosphere until the solution became a very viscous solution. The viscous solution was left static in an open atmosphere at room temperature for 12–16 h. The CS-MB hydrogel was taken from its mold and dried in open air until both faces of the molded hydrogel were sturdy enough to hold. It was then immersed in nanopure water for an hour. The water bath was changed every 20 min. The washed hydrogel was carefully placed flat on a rack and dried at room temperature until a dry hydrogel film was achieved. CS gel was fabricated using the same method except for no addition of Glu and 4-HC and was used as a reference.

### 2.3. Characterization

Shimadzu Inspirits Fourier-transform infrared (FT-IR) spectrophotometer equipped with ATR (ZnSe) was utilized to determine the chemical functional group of raw materials and CS-MB hydrogels. FEI Quanta-FEG 250 field-emission scanning electron microscope (SEM) with 1 nm resolution was used for morphological visualization and Oxford Aztec™ energy-dispersive spectrometer (EDS) coupled with SEM was used for surface elemental analysis.

### 2.4. Degree of Deacetylation of CS

The degree of deacetylation of CS that provides information on the free amines available for reaction was quantified with FT-IR using the mathematical relationship employed by J. Brugnerotto et al. [34], as shown in (1) and (2):

$$\textbf{\% Degree of acetylation} = \textbf{31.92}(\textbf{Abs}_{1320}/\textbf{Abs}_{1420}) - \textbf{12.20} \tag{1}$$

$$\textbf{\% Degree of deacetylation} = \textbf{100} - (\textbf{\% Degree of acetylation}) \tag{2}$$

where $Abs_{1320}$ and $Abs_{1420}$ are the absorbance values at 1320 and 1420 cm$^{-1}$, respectively.

### 2.5. Swelling Studies of CS-MB

The pH-dependent swelling studies of CS-MB hydrogel was conducted by immersing a known mass of hydrogel film (6.5 mg, thickness of ~0.05 mm, cut in rectangular shapes) in 5 mL of nanopure water or 0.1 M buffered solutions. The temperature-dependent swelling studies were conducted in the same manner but at a pH 4.0 buffered system at different temperature. The buffered solutions at pH 2.4, 4.0, 5.8, 7.4., 8.0, and 9.5 were prepared with HCl/KCl, acetic acid/sodium acetate, citric acid/sodium citrate, sodium phosphate monobasic/sodium phosphate dibasic, potassium phosphate monobasic/potassium phosphate dibasic, and glycine/NaOH, respectively. The hydrogel sample was taken out from

the water, dry-blotted carefully, and weighed. The swelling capacity at a certain time ($W_t$) was assessed by (3):

$$W_t = [(W_s - W_d)/W_d] \tag{3}$$

where $W_s$ and $W_d$ are the mass of the swollen and dry state hydrogel at a certain time t, respectively. The swelling capacity of CS-MB hydrogel was fitted using the Kelvin–Voigt and Maxwell models in (4) and (5), respectively:

$$S_{(t)} = W_t(1 - e^{-t/r}) \tag{4}$$

$$S_{(t)} = (rW_t)t + W_t \tag{5}$$

where t is the swelling time, $S_{(t)}$ is the swelling capacity at a certain time, $W_t$ is the swelling capacity at time t, and r is the rate parameter determined by the time it takes to reach 63% of the equilibrium swelling capacity.

### 2.6. Screening of $Cu^{2+}$ Uptake of CS-MB

The $Cu^{2+}$ solution used in this study was prepared by the dissolution of an appropriate mass of $CuCl_2 \cdot 2H_2O$ in nanopure water. Hydrogel film was cut into small rectangular pieces (~3–5 mg/piece) prior to use in $Cu^{2+}$ uptake studies. Pieces of hydrogel (25, 50, and 100 mg) were exposed to 20 mL of 120 mg/L solution of $Cu^{2+}$ adjusted to pH 4 using 0.01 M HCl/0.01 M NaOH in a plastic beaker in batch mode. The heterogenous mixture was shaken at 140 rpm at 22 °C using Innova 2300 platform shaker. Then, 1 mL of aliquot was taken at certain time interval. The aliquot was diluted with nanopure water to achieve a final volume of 10 mL. The treated water was then subjected to analyses by Shimadzu ICPE-9810 inductive coupled plasma emission spectrometer equipped with ASC-9800 auto sampler for the determination $Cu^{2+}$ concentration.

### 3. Results and Discussion

#### 3.1. Fabrication Process Design

The current method being introduced in the fabrication of chitosan-based hydrogel involves a three-component, single-step, and one-pot chemical crosslinking of CS, glutaraldehyde, and 4-HC by the C-Mannich reaction at ambient conditions, as illustrated in Scheme 1. There have been few reports on biopolymers grafted by the C-Mannich reaction linked by beta-amino alcohol [35] as well as other Mannich-type reactions [36,37]. Generally, all other aminomethylation reactions have also been incorporated for grafting chitosan hydrogel systems. Moreover, crosslinking moiety can have their distinct contributions to the overall chemical functionality and mechanical stability of hydrogel systems. In this work, beta-amino ketone crosslinked novel chitosan-based hydrogel was generated for the first time from an ambient condition one-pot C-Mannich reaction, whose properties and applications are yet to be established.

**Scheme 1.** Proposed CS hydrogelation by Mannich reaction.

This methodology advocates for a simple upstream processing of CS-MB for further utilization that attains an in situ hydrogelation and creation of a thin film upon drying that can be physically cut into shapes upon complete dehydration. Alternatively, one could choose to conduct the fabrication process in reaction wells of specific shape to achieve shape specifications in situ prior to complete drying, as described elsewhere [38]. Furthermore, few avenues were taken that led to the optimization of this process design, which is presented in Table 1.

**Table 1.** Optimization of CS-MB hydrogel fabrication. Basis conditions unless otherwise noted: 2.5% *w/v* CS in 3% *v/v* AcOH; 1 eq. CS, 20 eq. Glu, 40 eq. 4-HC; open atmosphere drying at 20–23 °C.

| Parameters | Outcome |
|---|---|
| CS Final Concentration<br><br>1.7% *w/v*<br>1.8% *w/v*<br>2.2% *w/v*<br>2.5% *w/v* | • The final concentration of CS was found to be relevant in obtaining a very viscous clear gel that can be used as an indicator of a successful hydrogelation.<br>• Final CS concentration < 2.1–2.2% *w/v* in the current study did not form a sturdy hydrogel but remained a flowing viscous white-yellow mix gel/slurry.<br>• Final CS concentration > 2.2% *w/v* resulted in a clear and almost non-flowing viscoelastic product prior to complete drying as shown in Figure 1A. |
| Molar equivalence<br><br>1 eq. CS only<br>1 eq. CS, 10 eq. Glu, 20 eq. 4-HC<br>1 eq. CS, 15 eq. Glu, 30 eq. 4-HC<br>1 eq. CS, 40 eq. Glu, 80 eq. 4-HC<br>1 eq. CS, 60 eq. Glu, 160 eq. 4-HC<br>1 eq. CS, 20 eq. Glu, 40 eq. 4-HC | • The molar equivalence between CS, Glu, and 4-HC has a significant role in the physical characteristic of the product.<br>• The partially dried hydrogel of the bare CS is very sticky and less elastic.<br>• The partially dried hydrogel of CS + Glu + 4-HC is a little turbid and more viscoelastic and can be taken out from the mold without easily breaking, specifically for the ratio of Glu ≥ 20 eq. as supported by previous literature as the optimized CS:Glu ratio [39].<br>• At 60 eq. addition of Glu, the mixture formed a non-viscoelastic hydrogel few minutes after the addition of the crosslinker without casting.<br>• The dried hydrogel obtained from 60 eq. addition of Glu retained the general shape of its mold and an observed significant increase in hardness, but low water swelling capability.<br>• It was generally observed in the present system that the higher the molar equivalence of Glu, the stiffer the dried product and the lower the swelling capacity, which can suggest high degree of crosslinking and increase in hydrophobicity, as portrayed in Figure 1B. |
| Linker solution<br><br>3% *v/v* acetic acid (AcOH)<br>100% Ethanol (EtOH)<br>8: 2 EtOH to 3% *v/v* AcOH | • The choice in the linker solution is to address the solubility of 4-HC; 40 eq. of 4-HC did not dissolve in 3% *v/v* AcOH, but completely soluble in EtOH.<br>• Using 100% EtOH created hydrogelation issues. A previous study proposed that CS solubility in 2% AcOH media is diminished by increasing EtOH concentration [40], decreasing the final CS concentration.<br>• The desired viscoelasticity of CS-MB and 4-HC solubility was achieved with 8:2 EtOH to 3% *v/v* AcOH. Visual indication of successful CS-MB hydrogelation is illustrated in Figure 1C. |

The molar equivalence of 1:2 between Glu and 4-HC was proposed based on the reaction mechanism expected from a typical acid-catalyzed C-Mannich reaction described by Cummings and Shelton [41], as simplified in Scheme 2. Briefly, the first step in the expected reaction mechanism is the formation of an electrophilic iminium (Schiff reaction) from the primary amine substituent of the CS and one aldehyde end of the Glu. In acidic media, 4-HC is known to undergo a keto-enol tautomerism. The carbon position 3 of the enol form of 4-HC is presented as a good nucleophile that will attack the electrophilic carbon of the iminium, forming the beta-amino ketone linkage. Further, the other aldehyde end of the Glu will undergo the same reaction mechanism with a different CS chain and another molecule of 4-HC, although there is uncertainty to whether both ends of the Glu

form iminium simultaneously or consecutively, and a kinetics study is outside the scope of this current work. The 4-HC was chosen as the nucleophile for this beta-amino ketone bridging because in our previous unpublished work on seeking small natural or bio-derived molecule nucleophiles for Mannich reaction, only 4-HC led to a complete reaction as a precipitate at ambient conditions with exceptional purity without the need of purification.

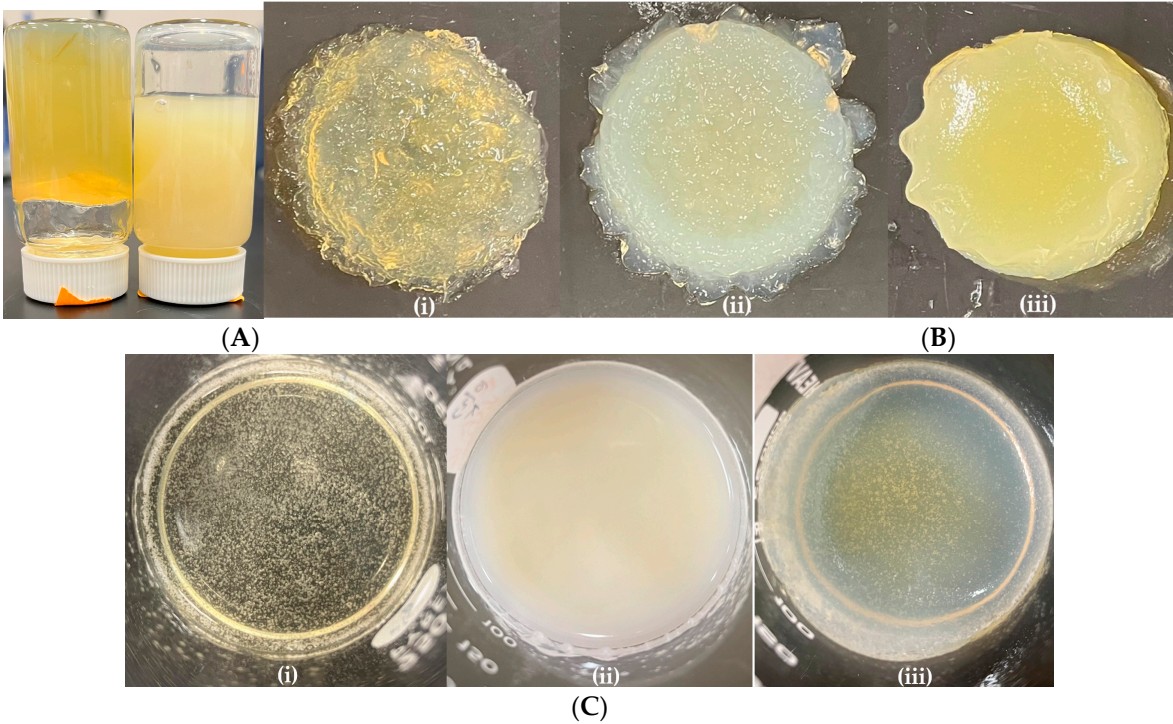

**Figure 1.** (**A**) Turbidity and viscosity of hydrogel solution at varying final CS concentrations post fabrication: (L) 2.1% *w/v*, (R) 1.7% *w/v*. (**B**) Hydrogel products after 16 h at varying component ratios: (i) 1 CS: 20 Glu: 20 4-HC, (ii) 1 CS: 30 Glu: 60 4-HC, (iii) 1 CS: 60 Glu: 120 4-HC. (**C**) Visual indications of successful CS-MB hydrogelation: (i) Pure CS gel in AcOH$_{(aq)}$, (ii) incomplete/unsuccessful, (iii) complete/successful.

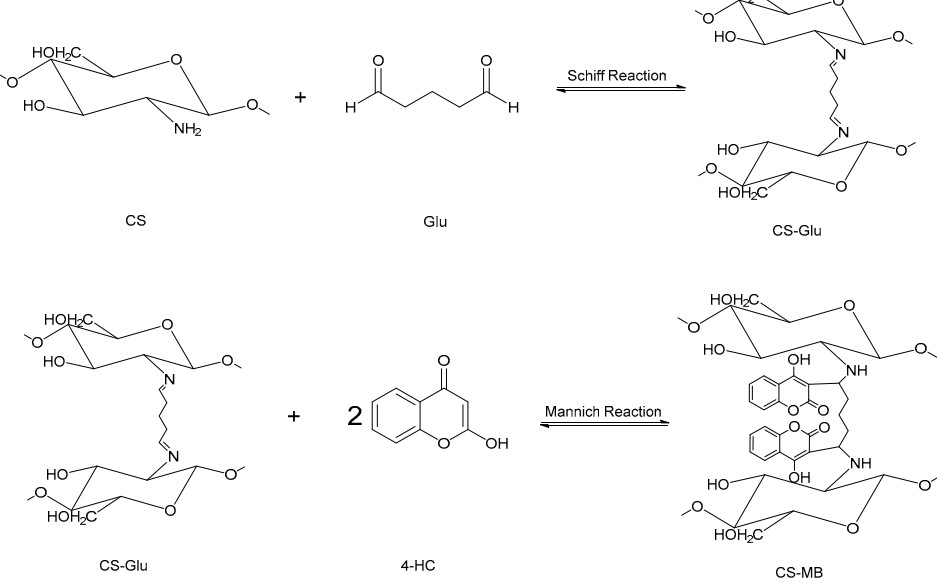

**Scheme 2.** Simplified mechanism of Mannich reaction (CS is represented as its glucosamine monomer).

A reaction was run with 1 eq. CS: 20 eq. Glu: 20 eq. 4-HC to provide evidence for the necessity of two molecules of 4-HC in every one molecule of Glu for a complete crosslinking by C-Mannich reaction to occur. After drying, the dried film was analyzed by FT-IR. The spectrum in Figure S1 shows no peaks around 1532 and 759 cm$^{-1}$ that would indicate aromatic C=C and C-H stretches from 4-HC, respectively, but instead, a weak to medium stretch between 1650–1700 cm$^{-1}$ was observed, which was previously assigned as the imine peak for a CS-Glu system [42]. This information suggests that in order for the C-Mannich reaction to dominate over the Schiff reaction to obtain a stable CS-MB hydrogel, there must be an appropriate molar equivalence of the nucleophile relative to the Glu. Otherwise, the equilibrium of the reaction will shift toward the reactants of the Mannich reaction, which would imply an increase in the CS-Glu system as dictated from Le Chatlier's principle, resulting in the potential collapse of the CS-MB to a CS-Glu system. Having relatively large excess of 4-HC results in an off-white opaque film. The opacity must be a result of the precipitation of 4-HC.

### 3.2. Physicochemical Properties

### 3.2.1. Degree of Deacetylation

The degree of acetylation was quantified in accordance with Equation (1). The absorbances at 1420 and 1320 cm$^{-1}$ were found to be 0.334 and 0.392, respectively. The ratio of the absorbance values of the peaks at 1420 and 1320 cm$^{-1}$ was quantified to be 1.174. By Equation (1), the degree of acetylation of the CS powder used in this study was found to be 25.26%; then, by Equation (2), the actual degree of deacetylation was determined to be 74.74%.

### 3.2.2. FT-IR Characteristic Peaks and Elemental Analysis

To determine the successful fabrication of the CS-MB hydrogel, FT-IR analysis was conducted, and the spectrum of CS-MB is shown in Figure 2. It is expected that for a successful crosslinking of CS with beta-amino ketone through 4-HC and glutaraldehyde, peaks corresponding to the C-H stretch and C=C vibrations due to the aromatic ring in 4-HC starting material must be observed in addition to the C-H vibrations characteristic of polysaccharides. The spectrum of the CS-MB hydrogel is presented in Figure 2. The peak at 3250 cm$^{-1}$ was assigned to an overlapped frequency of the O-H and N-H vibrations. The bands at 3090, 1652, 1613, and 759 cm$^{-1}$ were assigned to the aromatic C-H stretching, C=O lactone vibration, C=C ring distortion, and aromatic C-H out-of-plane bending, respectively, which are characteristic peaks of coumarins [43]. The peaks at 2923, 2874, and 1320 cm$^{-1}$ correspond to the symmetric C-H stretch, asymmetric C-H stretch, and the lactone C-O-C, while the peak at 1532 cm$^{-1}$ must be an overlap band of a second peak for the C=C ring distortions from 4-HC and N-H vibrations of CS. Finally, the bands at 1389, 1153, 1020, and 894 cm$^{-1}$ are characteristic peaks of CS, as shown in Figure S2. Additionally, elemental analysis of the CS hydrogel before and after functionalization was also conducted, as presented in Table 2. The increase in the C-atom constitutes the addition of Glu and 4-HC in the structure of CS film post-crosslinking, resulting in the decrease in the N-atom. To make it apparent, the C/N ratio was calculated for both CS film and CS-MB film. The C/N ratio for CS-MB is higher than that of the CS film, which verifies that successful fabrication of CS-MB, in addition to the FT-IR analysis conducted.

**Table 2.** Surface elemental analysis of CS-film and CS-MB by EDS (averaged between three different areas).

| Composition (%) | C | N | O | Others | C/N |
|---|---|---|---|---|---|
| CS film | 49.46 | 6.71 | 43.64 | 0.20 | 7.37 |
| CS-MB | 55.31 | 5.48 | 39.15 | 0.08 | 10.09 |

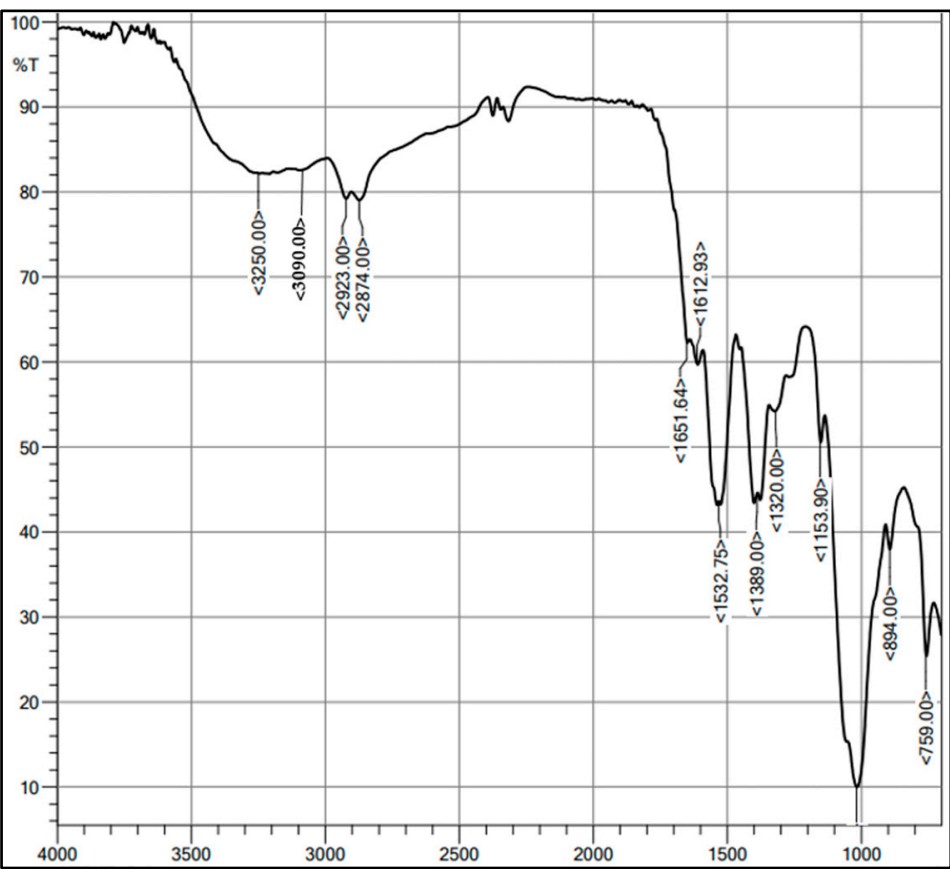

**Figure 2.** FT-IR of CS-MB.

3.2.3. Surface Characterization

The surface morphology was visualized with SEM, as shown in Figure 3, and the surface chemical composition was imaged with EDS, as shown in Figure 4. The CS film has a smooth surface prior to crosslinking. Upon crosslinking, roughness in the surface appears, but no obvious porosity is present. The absence of porosity in the SEM, however, must be due to the sample preparation for the SEM procedure, which completely dehydrates and freezes the sample for analysis. The combination of complete dehydration and freezing under vacuum can cause the CS-MB porosity to collapse. It is thought that the roughness in the surface of the CS-MB must be due to not only the presence of hydrophobic groups, but also to a remnant of the collapsed pores of the CS-MB. To test this hypothesis, the CS-MB was swollen with saline solution, surface-washed thoroughly, and air-dried prior to SEM analysis. Salts cannot be removed by lyophilization and will remain either trapped in the internal network structure or pores of the CS-MB or visibly precipitate in the surface upon dehydration. As evident from Figure 3iv–vi, the porosity of the CS-MB is most likely preserved by the efflux of NaCl to the surface during the freeze-drying process under vacuum, avoiding the total collapse of the pores from CS-MB. Furthermore, in terms of chemical composition at the surface, it is apparent from Figure 4 that there is almost complete uniformity except for the presence of some artifacts. These surface impurities are trapped air bubbles upon fabrication and during post-fabrication sample handling.

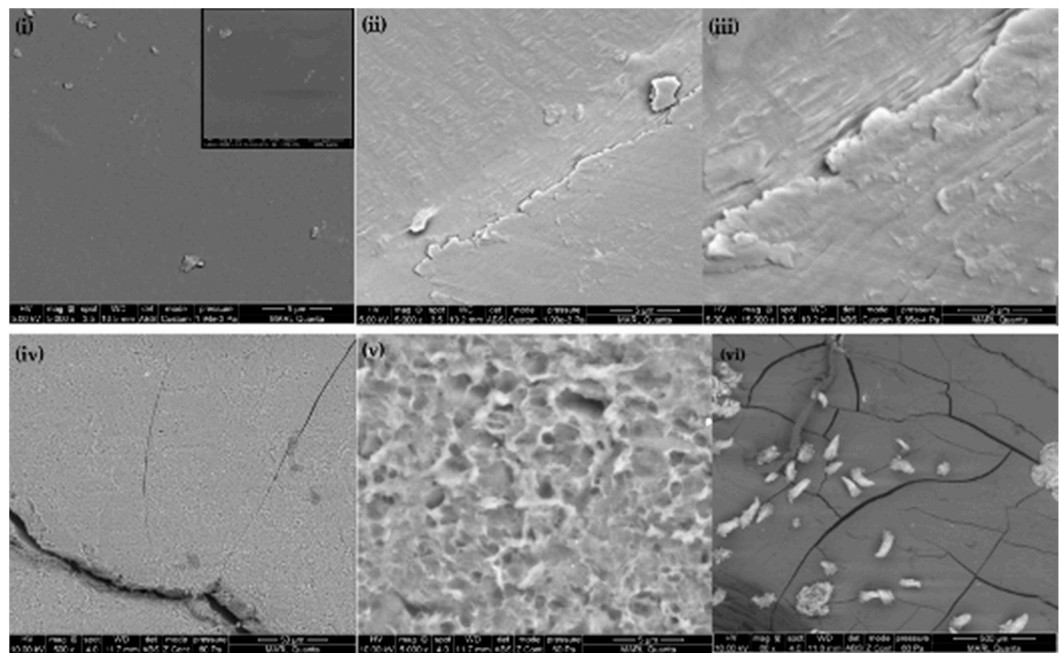

**Figure 3.** SEM images of CS-MB hydrogels: (**i**) CS film at 5000×; (**ii**) CS-MB at 5000×; (**iii**) CS-MB at 15,000×; (**iv**) CS-MB swollen in 5% *w/v* NaCl 500×; (**v**) CS-MB swollen in 5% *w/v* NaCl at 5000×; (**vi**) CS-MB swollen in 5% *w/v* NaCl at 60× with visible NaCl on the surface.

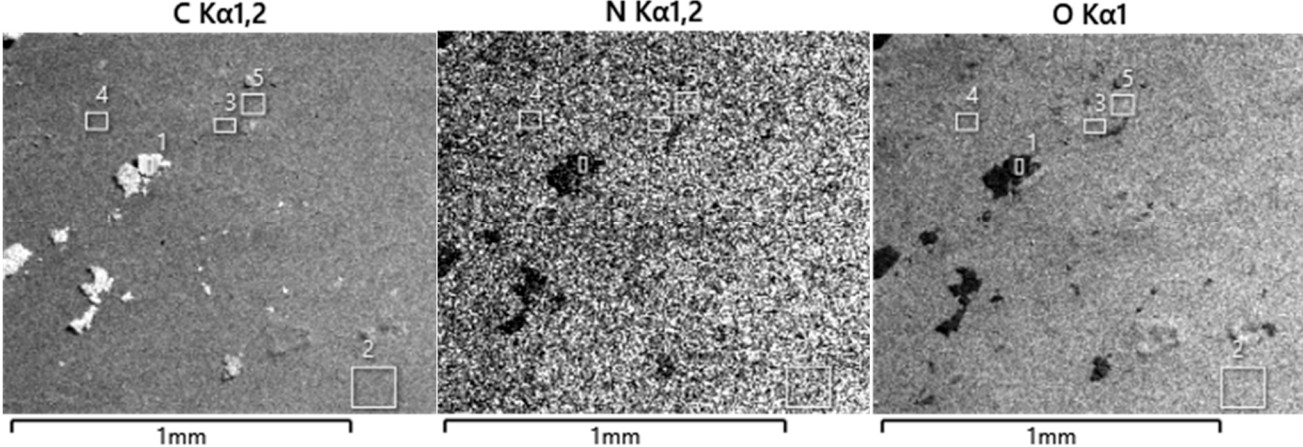

**Figure 4.** EDS image of CS-MB hydrogel. Conditions: Samples immobilized to carbon tape and coated with 5 nm of iridium for conductivity and then imaged at 5 kV.

### 3.2.4. Confirmation of CS-MB as a Hydrogel Material

Hydrogel is a polymer that can be distinguished from other classes of polymers by its ability to swell in water. The degree to which hydrogel swells is governed by the affinity of water molecules to the hydrogel system both within the hydrogel three-dimensional network and at the interface. The extent of swelling provides information on not only the swelling capacity of hydrogels, but is also a confirmation of the polymeric material's mechanical properties based on its swelling behavior. Figure 5 portrays fitting the experimental data of CS-MB swelling in nanopure water using two common models used to describe viscoelastic polymeric materials [44]. The CS-MB hydrogel fits the Kelvin–Voight model ($R^2$ = 0.977368075) better than the Maxwell model ($R^2$ = 0.695262657). Fitting the experimental data with a Kelvin–Voight model confirms that the CS-MB is a viscoelastic material as observed on its physical characteristics undergoing deformation. Generally, this implies that the CS-MB is an amorphous polymer that consists of stretchable crosslinkers, resulting in its viscous and elastic properties, respectively. The experimental data fitting to

the Kelvin–Voigt model alludes to the fact that the CS-MB hydrogel experiences a reversible but decreasing rate of deformation (strain from swelling) under constant stress. Eventually, it will reach an equilibrium deformation characterized by forming an asymptote toward the maximum swelling capacity.

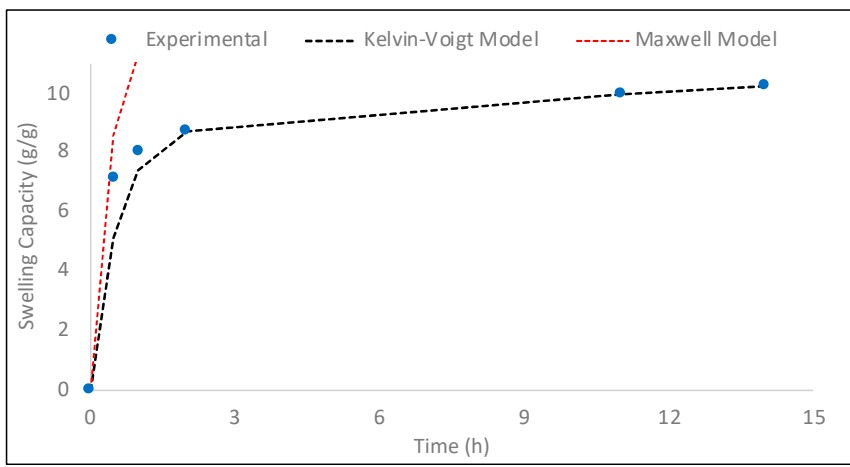

**Figure 5.** Fitting of experimental swelling capacity data with known model for viscoelastic polymeric materials. Conditions: 6.2 mg and thickness of 0.05 mm of hydrogel, 5 mL buffered solution at pH 4.0, and T = 22 °C.

### 3.2.5. Effects of pH of the Aqueous Medium on the Swelling Behavior of CS-MB

Swelling studies were conducted at different pH conditions to determine more information about the physicochemical characteristics of CS-MB hydrogel. Figure 6 illustrates that CS-MB presents a unique swelling behavior with respect to the pH of water. The swelling behavior of CS-MB at key pH is rationalized in Table S1. The bare CS film, meanwhile, is overswelled and disintegrated in the unbuffered distilled water. The swelling capacities of the bare CS and CS-MB in unbuffered distilled water after 4 min were 25.7 and 8.09 g/g, respectively. This difference in swelling capacity between the bare CS and CS-MB is a result of the hydrophobicity introduced in the CS-MB system upon crosslinking by C-Mannich reaction that consists of both the hydrophobic alkyl chains from Glu and the fused benzene ring in the 4-HC, significantly lowering the swelling capacity from its CS starting material. Compared to other CS hydrogel systems, swelling capacity continuously decreases with pH for two-component (CS + crosslinker) CS hydrogel systems [45–47], whereas multicomponent blended CS hydrogels [48,49] have their own unique swelling pattern depending on the different types of ionizable groups. The CS-MB system swelling behavior in response to pH is comparable to those reported for the latter, with one distinction: there is an inversion of trend in swelling capacity with time (for 5.5 < pH < 6.9 and pH > 8.1) that is not observed in other systems.

Additionally, to provide evidence on the stability of beta-amino ketone linkage under extreme pH conditions, the dried spent hydrogel pieces from the pH studies were analyzed by FT-IR to determine the chemical integrity of CS-MB presented in Figures S3–S8. The collapse of the beta-amino ketone linkage in the CS-MB hydrogel when exposed to different pH must be characterized by the absence of molecular vibrations and stretches corresponding to the aromatic ring in 4-HC, namely those at 2923, 1652, 1613, and 759 cm$^{-1}$. It is evident from the FT-IR spectra that these peaks, together with the polysaccharide characteristic bands at 1153, 1020, and 894 cm$^{-1}$, were preserved in the CS-MB exposed to extreme conditions, especially to acidic and basic environments where the imine linkage-based hydrogels and coumarin moiety, respectively, were known to hydrolyze. The only distinct difference was observed for the sample exposed to pH 8 and 10, as there is a significant blue-shifting of peaks corresponding to C=O lactone vibration and C=C ring distortion peaks. This could be a result of the metal countercations produced from the buffer that non-covalently interact with the 4-HC moiety.

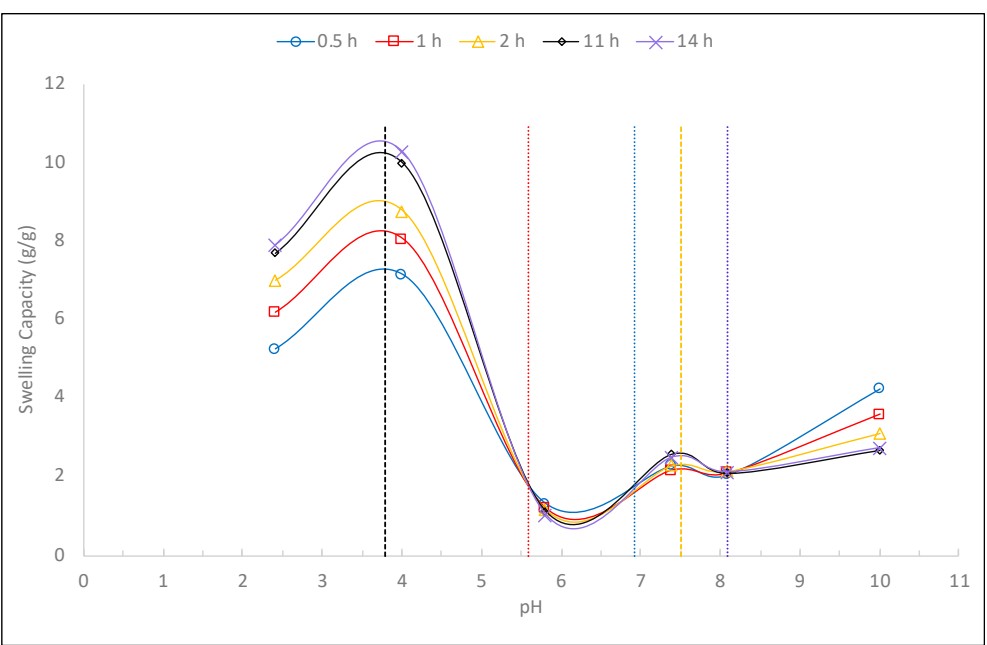

**Figure 6.** Effects of pH and swelling time on the swelling behavior of CS-MB. Conditions: 6.7 ± 0.3 mg and thickness of 0.05 mm of hydrogel, 5 mL buffered solution, and T = 22 °C.

### 3.2.6. Effects of Temperature on the Swelling Behavior of CS-MB

The effect of water temperature on swelling behavior of CS-MB in aqueous medium was also investigated, as presented in Figure 7. Generally, the swelling capacity improved as the temperature was raised from around 9 to about 55 °C, except for a drop in the water adsorption capacity that occurred around 34 °C. This occurrence is consistent regardless of the swelling conditions, where the error bars for each data point represent the standard deviation. The difference in the swelling measurements between duplicates can be accounted for due to the variation in the room temperature, humidity, and the amount of time the hydrogel is dry-blotted or sits in open air before weighing during the time of the data collection. It was observed that de-swelling of the hydrogel sample is sensitive to these factors.

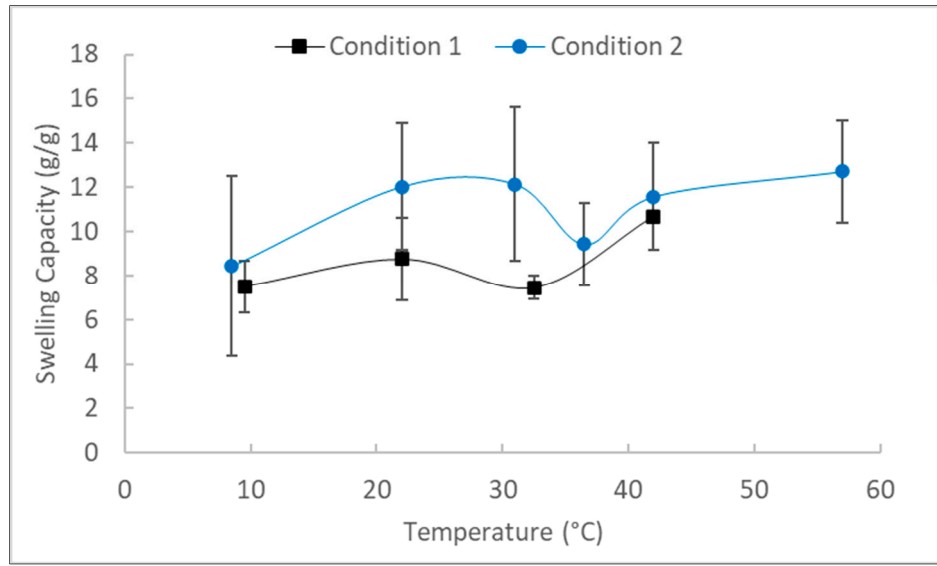

**Figure 7.** Effects of water temperature in the swelling capacity of CS-MB. Condition 1: 6.0 ± 2.0 mg CS-MB immersion in 5 mL unbuffered distilled water for 0.5 h. Condition 2: 4.5 ± 1.5 mg CS-MB, immersion in 5 mL pH 4 buffered solution for 11 h. The error bars represent standard deviation.

The drop in swelling capacity at a certain higher temperature than ambient temperature was observed previously on chitosan/PEG/PNIPAAm hydrogel films [50]. However, unlike the CS-MB in the current study, the swelling capacity of chitosan/PEG/PNIPAAm continued to decrease past this temperature, creating a sigmoidal curve. Freeze-dried chitosan hydrogels also exhibit an uncommon swelling behavior with temperature having a trough point in a U-shaped curve that defines the temperature at which the swelling capacity is the lowest [51]. Another example of a thermosensitive hydrogel system with a unique swelling behavior with temperature is the carrageenan-g-poly(aam-co-ia)/montmorillonite superabsorbent hydrogel composite that features a crest point as the temperature of maximal swelling of an inverse U-shaped graph [52]. The CS-MB offers a distinct swelling behavior with temperature in terms of swelling capacity pattern that no other hydrogel system has exhibited before, to the best of our knowledge.

This sudden drop in the swelling capacity defies the swelling behavior of thermosensitive chitosan hydrogel with a well-defined lower critical solution temperature (LCST), such as the chitosan/PEG/PNIPAAm hydrogel films. Both the concept of LCST and the volume phase transition temperature (VPTT) can explain this drop in swelling capacity around 34 °C relative to the subsequent temperatures. Briefly, LCST is the lower bound temperature, at which hydrogen bonding induced solvent–polymer interaction and polymer–polymer interaction are in competition [53]. Below the LCST, it was established that the hydrogen bonding–polymer interaction predominates, leading to a high degree of swelling or complete homogenous dissolution of the polymer in water [54]. Conversely, the polymer–polymer interactions prevail at temperatures above the LCST that results in polymer aggregation and less solubility in water and hence a low degree of swelling.

The VPTT, on the other hand, relates to the polymer swelling capacity and the equilibrium swelling capacity ($S_{eq}$) as a function of temperature, and the temperature that shows anomalous behavior during swelling is observed [55]. The VPTT of a polymer is near the defined LCST. In the CS-MB system, the VPTT can be approximated to be around 34 °C. This temperature is within the LCST range identified for modified-chitosan hydrogels at 32–42 °C [56]. The CS-MB system is composed of hydroxyl and amino hydrophilic groups as well as alkyl and benzyl hydrophobic groups. At the VPTT, it is expected that the alkyl and benzylic groups of the CS-MB are aggregated, blocking the accessibility of water molecules to approach the hydroxyl and amino groups for hydrogen-bonding interaction. This inaccessibility hinders the expansion of the CS-MB by volume shrinking at the molecular level. At 22 °C, the hydrophobic aggregation is diminished, and the CS-MB is in its expanded state, where it becomes highly accessible for water molecules to interact with hydrophilic groups, leading to a higher $S_{eq}$ than the VPTT. Above the VPTT, the hydrophobic interactions are still present, but they are stronger than at the VPTT. However, at higher temperature, water transport from the bulk solvent to the hydrogel is governed by convection, improving the $S_{eq}$ above the VPTT, as also observed by Goycoolea et al. in chitosan/PEG/PNIPAAm hydrogel film [51].

The swelling capacity observed for CS-MB equilibrated at around 10 °C dropped from 22 °C. Although there is no direct evidence, it was speculated that this behavior at low temperature is similar to the governing mechanisms in cold denaturation of proteins because, similar to proteins, CS-MB is amphoteric in nature. Proteins undergo cold denaturation when there is a significant hydration of hydrophobic side chains that disrupts the hydrogen bonding network in the system, leading to the following thermodynamic parameters: a negative or very minimal positive change in enthalpy, large value of change in heat capacity, and an overall increase in entropy driving the spontaneity of the process [57]. This mechanism is caused by both conformational changes of protein and reconfiguration of water molecule geometry that affects its degree of charge separation at low temperatures, as recently revisited by Khuri et al. [58]. The CS-MB at 10 °C is expected to be in its expanded state where the hydrophobic groups are well-exposed; thus, it is easily accessible to water molecules. This supposedly entropically favored occurrence dictates a stronger water–hydrophobic group interaction than that of the water–hydrophilic groups. As a result of the

entropically unfavorable interactions between water molecules and the hydrophilic group of the CS-MB, equilibrium swelling capacity at this temperature is relatively diminished.

### 3.3. Screening of $Cu^{2+}$ Removal with CS-MB

Our society has been challenged for many decades with declining water quality due to chemical pollutants that affect all aspects of water usage. Rapid growth of industrialization parallels the continuing contamination of water systems with heavy metals such as $Cu^{2+}$ from poor wastewater quality disposal from manufacturing plants that causes phytotoxicity and harm to agriculture biochemistry in the environment [59]. Additionally, leaching of copper pipes commonly utilized in residential water supply lines imposes tremendous health risks to consumers, as discussed elsewhere [60]. In terms of removal of $Cu^{2+}$ from water systems, natural chitosan is known to adsorb $Cu^{2+}$ efficiently, hence the subjected adsorbate in the CS-MB metal uptake studies.

The $Cu^{2+}$ uptake batch study was conducted at pH 4.0 and 22 °C, corresponding to the optimal equilibrium swelling capacity of the CS-MB portrayed in Figure S9. At pH 4, not only is this condition beneficial to the swelling capacity of CS-MB, but also, free $Cu^{2+}$ is the dominant copper species in this condition, as simulated from Visual MINTEQ 3.1 and presented in Table S2. CS-MB consists of several functional moieties that are capable of non-covalent interactions with metal ions, including hydroxyl (-OH), amine ($-NH_2$), and amide from the CS and the olefin, lactone ring, and benzene moiety from the 4-HC. CS-MB is a relatively new hydrogel system, and its metal complexing ability is still unexplored. This current work will screen the adsorption capability of CS-MB for the removal of $Cu^{2+}$ in aqueous systems.

The screening of $Cu^{2+}$ was conducted by determining the adsorption capacity of CS-MB toward $Cu^{2+}$ as a function of the CS-MB dosage. This experiment was conducted in duplicate, and the represented standard deviation error bars are provided in Figure S10. As shown in Figure 8, it is apparent that the adsorption capacity is diminished as adsorbent dosage is increased, which seems to contradict common sense, as an increase in adsorbent dosage also increases the total surface area for adsorption. However, in solid–liquid adsorption, aggregation of the adsorbent could occur that will decrease the exposed surface area for adsorption. Additionally, competitive adsorption and competing surface reaction (complexation, ion-exchange, absorption, etc.) could also play a role, leading to this occurrence. There are previously reported CS hydrogel systems that also observed similar trend. Few examples of these were compared in terms of adsorption capacity as a function of dosage reported in Table 3. As shown, although CS seems to show less superiority in terms of $Cu^{2+}$ adsorption performance per unit than most, the simple fabrication methodology of CS-MB is far more superior.

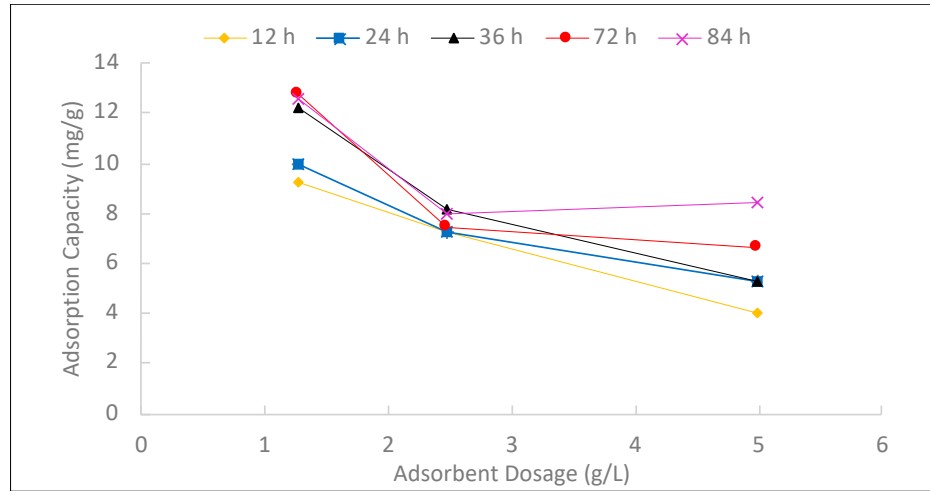

**Figure 8.** Screening of the adsorption capacity of CS-MB with $Cu^{2+}$ as a function of adsorbent dosage at different time intervals.

**Table 3.** Adsorption performance of modified CS hydrogel system.

| Hydrogel | Max Adsorption Dosage (g/L) | Max Adsorption Capacity (mg/g) | Reference |
|---|---|---|---|
| CS-MB film | 1.25 | 12 | This work |
| CS beads | 0.20 | 86.3 | [61] |
| CS-Glu beads | 10 | 2.3 | [62] |
| CS-aryl | 1.0 | 59.6 | [63] |
| CS-tannic acid | 0.7 | 11.04 | [64] |
| CS-thiosemicarbazide | 1.5 | 47.16 | [65] |
| CS-Diphenylcarbazide | 1.0 | 185.51 | [66] |

For the current system, there is no competitive adsorption; thus, we conducted FT-IR analysis, as summarized in Table 4, of the fresh CS-MB hydrogel and the Cu-spent hydrogel (Figure S11) to detect the adsorption sites, while Figure 9 shows the fresh and spent hydrogel in their dry and swollen states. In terms of adsorption sites, the possible covalent interactions between the metal and CS-MB are complexation and chelation with the functional groups that are Lewis bases (hydroxyl and amine), whereas the cation–π interaction is anticipated between the aromatic ring and the metal ions at the appropriate geometry. The adsorption of $Cu^{2+}$ in CS-MB is characterized by both IR inactivity of the aromatic C-H and asymmetric C-H stretching as well as by a significant blue-shifting of the C=O lactone vibration and C=C ring distortion peaks. The IR inactivity of the former may not provide reliable interpretation because the absence of bands in this region can sometimes be due to a poor physical transmittance of light through a sample that is too translucent or other physical factors. Conversely, shifting in wavenumber is well-accepted evidence of metal adsorption onto an organic functional group, for example, the cation–π interaction that distorts benzene ring symmetry as previously investigated [67]. This output provides verification that 4-HC moiety in the CS-MB plays a role in the adsorption of $Cu^{2+}$.

**Table 4.** Key FT-IR bands comparison between fresh and spent CS-MB.

| Assigned Group | Fresh CS-MB | Spent CS-MB |
|---|---|---|
| Aromatic C-H stretching | 3090 cm$^{-1}$ | IR inactive |
| Asymmetric C-H stretch | 2874 cm$^{-1}$ | IR inactive |
| C=O lactone vibration | 1652 cm$^{-1}$ | 1687 cm$^{-1}$ |
| C=C ring distortion | 1613 cm$^{-1}$ | 1648 cm$^{-1}$ |

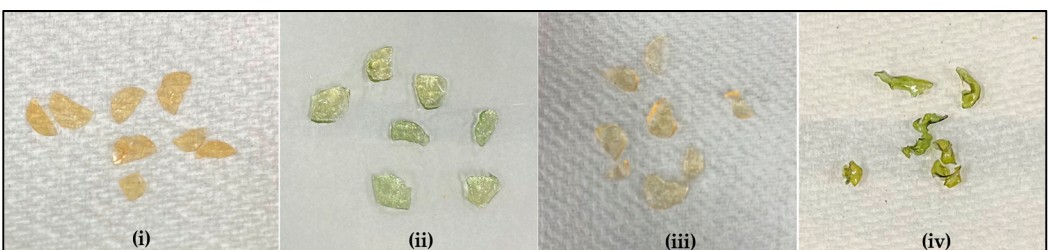

**Figure 9.** Fresh and $Cu^{2+}$ uptake spent hydrogels in their dry and swollen states: (**i**) fresh dry CS-MB; (**ii**) $Cu^{2+}$ uptake spent swollen CS-MB; (**iii**) fresh swollen CS-MB; (**iv**) $Cu^{2+}$ uptake spent dry CS-MB.

To provide verification that the changes in the molecular vibrations and stretching from FT-IR as well as the resulting color of the spent hydrogel are due to adsorbed $Cu^{2+}$, EDS surface analysis of both fresh and spent CS-MB was also pursued. The fresh CS-MB did not contain any $Cu^{2+}$, whereas $Cu^{2+}$ was detected in the spent CS-MB focused on different areas labeled, as in areas #1, #2, and #3 presented in Table S3. Furthermore, the EDS image of spent CS-MB, presented in Figure 10, was also scrutinized. Like the EDS image of the fresh CS-MB, the spent CS-MB also contained artifacts that are not local to the composition of either the CS-MB or the $Cu^{2+}$ water sample. Nonetheless, the uniform distribution of

$Cu^{2+}$ on the surface of CS-MB means that there is no localization or aggregation of $Cu^{2+}$, but it adsorbed rather evenly at the surface, which is feasible for this application.

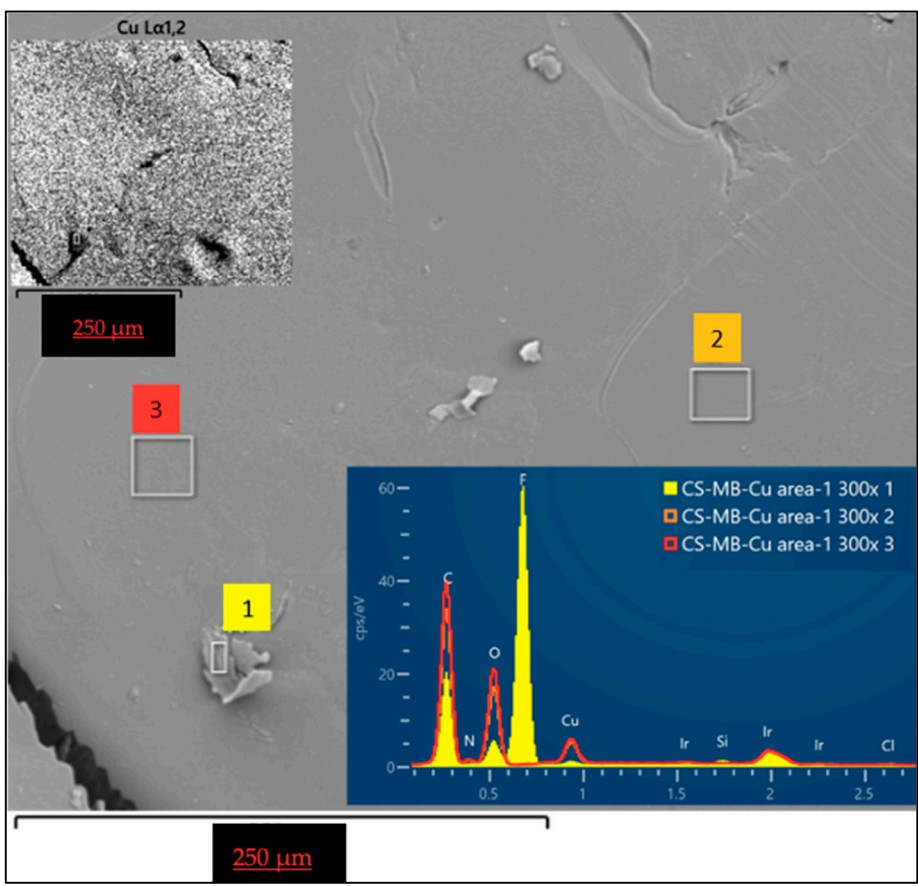

**Figure 10.** EDS image of spent CS-MB. Area 1 is identified as an artifact in this EDS image.

## 4. Conclusions

To summarize the key findings of this work:

(i) A novel beta-amino ketone crosslinked chitosan (CS-MB) was fabricated by a single-step and one pot C-Mannich reaction.

(ii) The equivalence ratio between CS, Glu, and 4-HC is crucial to achieve successful crosslinking with a feasible physical property.

(iii) The CS-MB hydrogel showed a unique swelling behavior in response to pH and temperature in which an inversion of swelling capacity with time was observed in response to pH, whereas there is a decrease in the swelling capacity below the VPTT while an increase in the swelling capacity was observed at temperatures above the VPTT.

(iv) The CS-MB exhibits adsorption activity against $Cu^{2+}$ in aqueous medium.

(v) The proposed adsorption sites for $Cu^{2+}$ in CS-MB are from the 4-HC moiety: benzene ring by cation–π interaction and the ester group of the lactone ring by electrostatic interaction.

**Supplementary Materials:** The following supporting information can be downloaded at: https://www.mdpi.com/article/10.3390/pr11020354/s1, Figure S1: FT-IR of CS-MB; Figure S2: FT-IR of CS hydrogel film; Figure S3: FT-IR of fresh CS-MB; Figure S4: FT-IR of exhausted CS-MB swelled at pH 2.4; Figure S5: FT-IR of exhausted CS-MB swelled at pH 4.0; Figure S6: FT-IR of exhausted CS-MB swelled at pH 7.4; Figure S7: FT-IR of exhausted CS-MB swelled at pH 8.1; Figure S8: FT-IR of exhausted CS-MB swelled at pH 10.0; Figure S9: Influence of pH and temperature of water on the equilibrium swelling capacity of CS-MB; Figure S10: Adsorption equilibrium studies in batch

mode with varying Cs-MB dosage; Figure S11: FT-IR of spent CS-MB; Table S1: pH sensitivity of CS-MB at 22 °C at normal atmospheric pressure; Table S2: Copper speciation in AcOH/NaOAC buffered system simulated from Visual MINTEQ; Table S3: EDC elemental analysis (wt%) of fresh and spent CS-MB.

**Author Contributions:** Conceptualization, J.R.A.R.; Formal analysis, J.R.A.R.; Investigation, J.R.A.R.; Methodology, J.R.A.R.; Project administration, J.R.A.R.; Resources, S.K.O.; Supervision, S.K.O.; Validation, S.K.O.; Writing—original draft, J.R.A.R.; Writing—review and editing, S.K.O. All authors have read and agreed to the published version of the manuscript.

**Funding:** This research received no external funding.

**Data Availability Statement:** Not applicable.

**Conflicts of Interest:** The authors declare no conflict of interest.

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
