# Peer review of "Single-Step Fabrication of a Dual-Sensitive Chitosan Hydrogel by C-Mannich Reaction: Synthesis, Physicochemical Properties, and Screening of its Cu2+ Uptake"

_processes, doi:10.3390/pr11020354_

Round 1

Reviewer 1 Report

The authors address an important topic in relation to the production of materials that absorb pollutants and metals, in this case copper. The text is well written. I indicate publication after the authors have considered some of the suggestions described below.

-         The authors explain the chose of copper in terms of efficiency of absorption by chitosan. However, they do not mention possible effluents containing copper and environmental risks of the Cu2+. Please add a paragraph to highlight this information.

-        I think Table 1 and 3 would look better in the supplementary material.

-        What chemical compound was used to produce 120 mg/L solution of Cu2+?

-        I did not understand the relationship of work with waste materials, since the article produced the CS-MB synthetically. Why is this covered in the introduction? Please explain further or remove this.

Reviewer 2 Report

The manuscript entitled  “Single step fabrication of a dual-sensitive chitosan hydrogel by C-Mannich reaction: Synthesis, physicochemical properties, and screening of its Cu2+ uptake” written by Romal and Ong describe a single-step procedure for hydrogel chitosan based preparation. The Data and the results are presented clear but in my opinion the discussion can be improved by included data, advantages compared to the previously obtained chitosan-derivatives gels by Mannich strategy. For example the work described by Chalitangkoon must be considered.  https://doi.org/10.1016/j.carbpol.2019.115049

or a recent review https://doi.org/10.1002/agt2.21 .

Overall the work is interesting and with some improvements can be suitable for publication in Processes Journal.
